# Pathogenic Aspects and Therapeutic Avenues of Autophagy in Parkinson’s Disease

**DOI:** 10.3390/cells12040621

**Published:** 2023-02-15

**Authors:** Rémi Kinet, Benjamin Dehay

**Affiliations:** Univ. de Bordeaux, CNRS, IMN, UMR 5293, F-33000 Bordeaux, France

**Keywords:** Parkinson’s disease, autophagy, genetic, lysosome, drugs, therapeutics

## Abstract

The progressive aging of the population and the fact that Parkinson’s disease currently does not have any curative treatment turn out to be essential issues in the following years, where research has to play a critical role in developing therapy. Understanding this neurodegenerative disorder keeps advancing, proving the discovery of new pathogenesis-related genes through genome-wide association analysis. Furthermore, the understanding of its close link with the disruption of autophagy mechanisms in the last few years permits the elaboration of new animal models mimicking, through multiple pathways, different aspects of autophagic dysregulation, with the presence of pathological hallmarks, in brain regions affected by Parkinson’s disease. The synergic advances in these fields permit the elaboration of multiple therapeutic strategies for restoring autophagy activity. This review discusses the features of Parkinson’s disease, the autophagy mechanisms and their involvement in pathogenesis, and the current methods to correct this cellular pathway, from the development of animal models to the potentially curative treatments in the preclinical and clinical phase studies, which are the hope for patients who do not currently have any curative treatment.

## 1. Introduction

The threat of Parkinson’s disease (PD) has been studied since its first description by James Parkinson at the beginning of the 19th century [1]. Still, the global understanding of this neurodegenerative pathology has increased exponentially since the end of the 20th century. Indeed, a better understanding of symptoms from dopaminergic (DA) deficiency [2] allows for the development of the first symptomatic treatment with Levodopa, the precursor to dopamine, which is still one of the most effective drugs for Parkinson’s disease (PD) but is associated with significant side effects such as dyskinesia and habituation [3]. Furthermore, the discovery of the α-synuclein (α-syn) role [4] and a genetic role in pathogenesis by Polymeropoulos and colleagues [5] allows for the emergence of new fields of study, ranging from animal modeling to α-syn-based therapeutics.

Nowadays, the number of genes implicated is over 25 (for instance, the *TMEM175* gene emerged in 2022 as a gene involved in PD); further understanding of the pathology and its link with the cellular clearance mechanism of autophagy are highlighted. Other drugs appeared, like monoamine oxidase type B [6] or catechol-O-methyl transferase [7] inhibitors. For example, the development of deep-brain stimulation [8] permits better nursing of PD patients, but these symptomatic treatments are not curative and are still accompanied by possible side effects.

In this review, we discuss the new insight into PD with the appearance of new genetic risk factors and the autophagy process and its implication in neurons and specifically in the pathology, making this mechanism an interesting way to develop curative treatments, some of which are presently in preclinical and clinical phases and represent an important hope for the cure of PD.

## 2. Parkinson’s Disease

Epidemiologically and globally, the burden of PD is approaching a threefold increase in 26 years, from 2.5 million patients in 1990 to 6.1 million patients in 2016. Over the next 30 years, the number of PD patients should reach more than 12 million worldwide by about 2050 [9,10,11]. Clinically, PD is characterized partly by its non-motor symptoms, such as constipation, sleep disorder, depression, pain, or also dementia [12]. However, it is primarily known for its motor symptoms, including bradykinesia, body rigidity, dysphagia, and tremor [13,14]. The loss of several neuronal populations characterizes PD. Still, the most characterized one is the extensive cell loss of DA neurons in the *substantia nigra* (SN), an essential input in the regulation of the basal ganglia motor loop through the nigrostriatal pathway [15,16].

Along with this degeneration, the pathological hallmark of PD is the presence of intraneuronal proteinaceous cytoplasmic inclusions, named Lewy bodies (LB), that invade the whole nervous system as the disease progresses [17]. Several fundamental discoveries have strongly implicated the protein α-syn in the pathogenesis of both familial and sporadic forms of PD (for in-depth review [18]). Other proteins [19], lipids [20], and organelles [21,22] also accumulate in LB [19], but α-syn is the main protein component [4].

PD pathogenesis can occur from environmental factors such as pesticide exposure, head trauma, or aging [23] and can also be the result of genetic mutations, with an increasing list of genes and mutations related to familial PD, comprising autosomal dominant forms like *SNCA*, *LRRK2*, or *VPS35* and autosomal recessive genes like *PRKN*, *PINK1*, *PARK7*, and *DJ1* mutations (Table 1).

## 3. Autophagy Mechanism

Autophagy is one of the two cellular processes in charge of cell clearance and maintaining cellular homeostasis. Disruption of autophagy is linked to multiple cellular malfunctions, starting with an accumulation of non-degraded elements [63], enhancement of reactive oxygen species (ROS) production [64], and neuronal bioenergetic imbalance [65]. Autophagy dysfunction thus appears as an essential component of PD pathogenesis [66,67]. Autophagy can be divided into three main pathways: macroautophagy (MA), chaperone-mediated autophagy (CMA), and microautophagy (Figure 1). Having all their particularities, they share the lysosome as a common organelle, permitting the elements’ degradation and recycling of their essential components. Lysosomes are tiny intracellular organelles with an average diameter of 500 nm [68] and contain over 50 different hydrolases [69], which are activated by the low lumen pH of the lysosome, which is around 4.5 [70]. This acidic pH, characteristic of a functional lysosome, is due to multiple membranous proton pumps, part of the more than 700 referenced lysosomal membrane proteins [71].

### 3.1. Macroautophagy

Macroautophagy (MA) is an autophagy-lysosomal pathway (ALP) mechanism that can be specific or not and is implicated in many cellular processes, like central nervous system development [72,73] and maintenance [74]. This mechanism for delivering cellular components to lysosomes for degradation is composed of multiple steps. 

First, the initiation phase permits phagophore production and is negatively regulated by the mechanistic target of rapamycin (mTOR) complex 1 (mTORC1). Formation of this lipidic double membrane, assembled mainly from the endoplasmic reticulum and Golgi membranes [75,76,77,78], is triggered by the ULK1 (unc-51-like kinase 1) complex formed with ATG (autophagy-related protein) 13, ATG101, and FIP200 (focal adhesion kinase family interacting protein of 200 kDa) [79,80,81,82]. The ULK1 complex permits activation of a second kinase complex composed of Beclin-1, ATG14, and vacuolar protein sorting (VPS) 15 and 34, generating a cascade of reactions resulting in the formation and expansion of the phagophore at the autophagy initiation site [83]. Selective MA uses autophagy receptors to initiate the formation of the phagophore, like phosphorylation, ubiquitination, acetylation, and oligomerization [84], which are recognized by phagophore proteins ATG8, LC3, and GABARAP [85]. Once closed, the now-called autophagosome can enter a maturation phase, fusing with endosomes to add new material to be degraded [86]. Finally, the autophagosome fuses with lysosomes via GTPases, lipids, and SNARE regulation [87], allowing content degradation.

### 3.2. Chaperone-Mediated Autophagy

CMA is a specific ALP process with a role in cell metabolism [88] and cycle [89,90]. CMA uses a recognition complex composed of the cytosolic Hsc70 protein (heat shock cognate 70 kDa) and co-chaperones that bind to a pentapeptide CMA-specific recognition motif (the KFERQ motif). This motif is composed of a succession of charged residues, hydrophobic residues, and glutamine. It can be formed and exposed through the 3D folding of the protein [91] or by acetylation [92] and phosphorylation [93] post-transcriptional modifications. Once the Hcp70 complex recognizes the KFERQ-like motif, it translocates and unfolds the substrate needed to be degraded. Finally, the chaperone-substrate complex binds to the monomeric lysosomal-associated membrane protein-2A (LAMP-2A), which then forms a multimeric complex to permit the lysosomal entry of the substrate to allow proteasomal degradation.

### 3.3. Endosomal Microautophagy

The lysosome can also directly phagocytose and degrade different elements like mitochondria [94,95], peroxisomes [96], part of the endoplasmic reticulum [97], and the nucleus [98]. This process, known as “microautophagy”, can be selective or non-selective, and it uses invagination or protrusion of the lysosomal membrane to capture elements that need to be degraded. LC3, ESCRT (Endosomal Sorting Complex Required for Transport), and the ATPase VPS4A are involved in the invagination microautophagy mechanism [97]. Still, they are also implicated in late endosome (LE) membrane modulation in endosomal microautophagy (eMI). This process, similar to microautophagy, uses a late endosome as a degradative compartment. The substrate uptake can be direct or selective with Hsc70 complex binding to LE membrane phosphatidyl serine before phagocyted in the lumen in ESCRT-complex dependent manner [99]. eMI activation is enhanced in oxidative or genotoxic stress conditions and results in the diminution of MA activity [100].

## 4. Autophagy Disruption in Parkinson’s Disease

Many neurodegenerative diseases, including PD, are linked to autophagy defects by different pathways. Herein, we focus on some of the most prevalent PD-related mutations causing autophagy phenotypes.

### 4.1. Genetic Implication in Autophagy Dysfunction

Autophagy impairment can result from genetic mutations. Constitutive MA is essential for neuronal survival, as its selective genetic inactivation in neurons leads to the formation of ubiquitinated intracellular inclusions and neuron cell loss in mutant mice [101,102]. ALP dysfunctions are caused by multiple genes in Parkinson’s disease (Table 1). Human genetics studies indicated that lysosomal dysfunction may play an essential role in the pathogenesis of PD; a recent genome-wide association study (GWAS) discovered a significant burden of rare, likely damaging LSD gene variants in association with PD risk, such that 56% of PD cases have at least one putatively damaging variant in an LSD gene [103]. Additional studies confirmed previous results, identifying 18 and 6 new loci associated with PD [104,105,106]. At least 11 of these loci are either directly or indirectly (i.e., disruptible) linked to the ALP, implying that lysosomal dysfunction may play a primary pathogenic role in Parkinson’s disease.

The MA mechanism dysfunction, caused by different gene mutations, can be linked with PD, with *SNCA* gene mutations being the first reported cause of PD cases, with primary evidence in 1997 and the last discovery in 2021. The eight-point mutations in *SNCA* known are A30G [107], A30P [108], E46K [109], H50Q [110,111], G51D [112], A53E [113], A53T [5], and A53V [114]. These multiple point mutations and the multiplication of this gene [115] are linked to different phenotypes of PD patients according to the mutations, from typical pathology to early-onset dementia, hallucinations, myoclonus, or pyramidal signs. But *SNCA*-PD patients all share usual hallmarks such as LB presence, neuronal depletion, and autophagic flux problems in MA and CMA mechanisms [116].

The mitophagy subtype of the MA mechanism can be affected explicitly by PD-related mutations. PTEN-induced putative kinase 1 (PINK1) and Parkin, respectively encoded by *PINK1* and *PRKN* genes, are two proteins interacting in the outer mitochondrial membrane and ubiquitinating it when the organelle is damaged, permitting its selective elimination [117]. *PINK1* or *PRKN* mutations result in the difficulty of mitochondrial elimination by disrupting the mitophagy initiation step, leading to their cellular accumulation [118] and causing autosomal recessive juvenile Parkinsonism [119,120]. Mutations of the *PARK7* gene encoding DJ1 are related to early-onset PD due to its interactions with PINK1 and the increased action of Parkin in wild-type conditions. The DJ1 inactivation induces a reduced steady-state level of PINK1, resulting in mitochondrial dysfunction [121].

*LLRK2* mutations are the most common genetic cause of late-onset PD [25,122], contributing to specific MA and CMA dysfunction. Over 50 mutations of this gene lead to dysregulation of cytosolic LRRK2 (leucine-rich repeat protein kinase-2) activity, which affects the endocytic process of a synaptic vesicle or lysosomal maintenance [123]. Significant outcomes of *LRRK2* mutations are an increase in lysosomal impairment and lysophagy due to LRRK2 enhanced activity [124], α-syn aggregation [125], autophagic vacuoles accumulation [26], and alteration of endolysosomal trafficking [126]. Heat shock proteins and co-chaperone mutations, playing an essential role in the CMA process, also dysregulate the machinery, like for *DNAJC6* encoding Hsp40. This co-chaperone regulating Hsc70 complex recruitment leads to ALP impairment in *DNAJC6* mutations [127].

The final and most common element of the ALP process, the lysosome, is also the target of different mutations, possibly modifying lysosomal pH or enzymatic activity and making it inactive. Glucocerebrosidase (GCase) is a lysosomal sphingolipid degrading enzyme, permitting the degradation of glucocerebroside in the lysosome. Mutations of *GBA* causing GCase lack leads to accumulation of substrate in the lysosomal lumen and α-syn aggregation [128]. ATP13A2 is another lysosomal protein that can be responsible for lysosomal malfunction. This metal cation transporter regulates lysosomal acidity and homeostasis by carrying Fe^3+^, Mn^2+^, Zn^2+^, and Ca^2+^. ATP13A2 and its gene *PARK9* impairments are linked to a decrease in clearance in the lysosome due to significant alterations such as acidification malfunction and reduction of the proteolytic enzymatic mechanism [38]. Recently, the TMEM175 proton leak channel of lysosomes and endosomes regulating lumen pH through a negative feedback mechanism has been identified as a PD risk [129]. Indeed, this channel regulating lysosomal functions through proton-activating [62], or growth-factor activating manner [130], once mutated in its pore or luminal loops regions, decreases its H^+^ permeability, diminishing lysosomal pH below the physiological 4.5 value and predisposing neurons to stress-induced damage and accelerates the accumulation of pathological α-syn because of the inactivation of enzymes [130,131], resulting in apoptosis promotion and aggravated PD symptoms [132]. In addition to lysosomal impairments caused by TMEM175 deficiency, mitochondrial functions also appear degraded [61]. Firstly, as observed in other synucleinopathies [133], variants of *TMEM175* were reported in 2023 in Italian PD patients [134].

### 4.2. Rodent Models Linked to ALP Dysfunctions

Different animal models are used in literature to understand better the ALP mechanisms and test strategies for restoring cellular clearance; herein, we discuss some of the novels or optimized rodent models based on ALP impairment involved in PD, proposed since 2018. ALP endosomal formation through the clathrin protein process is impaired by *SYNJ1* mutations encoding synaptojanin-1, an auxilin-like protein. Heterozygous deletion of *SYNJ1* in mice displays an age-dependent motor dysfunction phenotype in the rotarod test, with the histological observation of α-syn accumulation in SN and other brain regions, autophagy deficit, and DA neurodegeneration at 18 months old [135]. In 2021, the refinement of a mouse model with conditional knockout of autophagy-related gene 5 (*Atg5*) [136] was generated from a *Lyz2*^Cre^ (lysosome 2 Cre) strain expressing Cre recombinase in myeloid cells, and *Atg5*^f/f^ mice permitted *Atg5* silencing only in Cre-positive cells. The silencing of this protein involved in the autophagosome formation and maturation processes permits this mouse strain to observe down-regulated autophagy activity, neuroinflammation, and neurodegeneration with approximately 50% fewer DA neurons in the SN and exacerbated locomotor deficits [137]. The retromer complex also maintains endosomal functions, regulating the recycling of cargo proteins, and its subunit VPS35 is associated with AD forms of PD. The conditional *VPS35* D620N knock-in (KI) model obtained from *VPS35* D620N KI mice characterized by Cataldi et al. [138] crossed with mice from the Sox2-Cre-delete line results in motor deficiency after 14 months in the open field and narrow beam tests, along with a significant decrease of DA neurons in the SN and fibers in the striatum compared to control. This neurodegeneration is accompanied by a LAMP2A level decrease, lysosomal accumulation, and twice as much lipofuscin in the SN, suggesting autophagic dysfunction in a PD-like manner [139]. Reproducing PD mutations clinically observed in an animal model was also used in LRRK2 G2019S mice strain expressing the mutated protein, only in DA neurons through regulation of *LRRK2*-mutated gene by TH promotor, refining a 2011 model [140]. This model shows neurodegeneration of DA neurons and a significantly higher presence of an insoluble fraction of p129S α-syn at 15 and 24 months old in the striatum and the ventral midbrain Indeed, overexpression of the LRRK2 G2019S mouse model combined with injection of the MPTP neurotoxin resulted in severe motor impairment, selective loss of DA neurons, and increased astrocyte activation, indicating that the combination had a synergistic effect [141].

Causing lysosomal defects to mimic ALP-PD-related dysfunction is a strategy used in animal models, like with the recently described PD-related gene *TMEM175* KO mouse models. This model, obtained through CRISPR-Cas9 methods, leads to a decreased lysosomal pH at 4, reducing cathepsin B and D activity and a significative increase of phosphorylated α-syn presence compared to control mice. The accelerated fusion between autophagosome and lysosome was also reported, along with an accumulation of undigested autophagosomes in *TMEM175* KO models [130,131].

Another model of overexpression targeting the lysosomal dysfunction of human tyrosinase via intracerebral injection of a viral vector in the SN of both rats and mice permitted an age-dependent production and accumulation of neuromelanin in DA neurons, progressively occupying all the cytoplasmic space of the cell. Due to the insoluble property of neuromelanin, which makes it non-degradable, this model shows motor impairment in the contralateral side from 2 months after injection and leads to the observation of 6 times fewer DA neurons in the ipsilateral SN 24 months after injection, intracellular inclusion body formation, and lysosomal and autophagy dysfunction in rats [142].

The increasing number of animal models linked to ALP dysfunction that mimic the PD phenotype is a key step in understanding this pathology and developing and testing possible future treatments. In addition to these latest findings, it is important to note the usefulness of non-animal models, particularly brain organoids, in the first steps of study due to their capacity to generate complex and multicellular systems from patients’ pluripotent stem cells [143].

## 5. Autophagy-Related Therapies for PD

We have reported here that ALP impairment plays a pivotal role in different aspects of PD pathophysiology. According to these new findings, therapeutic modulation of autophagy in the context of PD and lysosomal-related pathologies paves the way for the development of a panel of therapeutic strategies (Figure 2).

### 5.1. Pharmacological Treatments

#### 5.1.1. mTOR-Dependent Drugs

The mechanistic target of rapamycin (mTOR) is a type of protein kinase involved in two complexes, the mTOR complex 2 regulating cell growth, proliferation, and protein synthesis, and the mTOR complex 1 (mTORC1) downregulating MA initiation, making it a therapeutic target objective. Early evidence suggests that rapamycin and derivative treatment reduces neurodegeneration and improves motor capacities and non-motor behavior in neurotoxin-MPTP or overexpressing α-syn mice models [144,145,146]. Other drugs are in the preclinical study phase to modulate the activity of mTOR and then increase autophagic activity. Caffeine also acts as an mTOR inhibitor, increases the number of autophagosomes, and reduces apoptosis, reestablishing autophagy activity [147,148]. Similarly, curcumin inhibits mTOR activity and permits autophagy enhancement [149]. AZD8055 shows interesting results as an mTOR inhibitor by increasing autophagy flux, the number of lysosomes, and acidified autolysosomes [150].

Similarly, corynoxine small-molecule was shown in 2021 to have neuroprotective effects, enhancing motor performances and decreasing α-syn aggregates in neurotoxic rotenone-induced mice PD-model [151]. Recently, small-molecule pyrazole derivatives invented by Kim and colleagues show remarkable mTORC1 inhibitory activity over 85% below 220 nM treatment without affecting mTORC2. Evaluated in mice, these molecules easily cross the blood-brain barrier (BBB) and decrease amyloid plaques, the Alzheimer’s disease hallmark [152]. Finally, new evidence for piperine as an autophagy enhancer emerged [153]. Its action to increase protein phosphatase 2A activity results in mTOR inhibition. It restores autophagy activity in a rotenone-induced PD mouse model [154] and attenuates olfactory and delayed motor deficits in transgenic mice overexpressing human *SNCA* [153]. The challenges of these treatments are to enhance autophagic flux targeting mTORC1 while avoiding disrupting cell survival, growth, or proliferation, in which mTORC2 plays a role [155], and find a good therapeutic dosage to induce average autophagic flux.

#### 5.1.2. mTOR-Independent Drugs

Some components use different targets than mTOR to restore ALP processes. Multiple autophagic regulators are raised as an attractive potential target through modulation of these protein levels, as further described in Scrivo et al. [156]. Beclin-1 involves the phagophore nucleation/elongation process and has multiple known enhancer drugs. For example, KYP-2047, a prolyl oligopeptidase inhibitor, induces expression of Beclin-1 and is associated with increased α-syn fibril degradation, amelioration of autophagy, resulting in better cell viability in vitro [157,158], but also in *SNCA*-A30P transgenic mouse model in which is the increase of MA and striatal dopamine levels with a 28-days treatment was reported [159]. Glycyrrhizin inhibits the activity of a Beclin-1 inhibitor and acts as an autophagy enhancer by multiple pathways: it decreases α-syn and glucocorticoid levels and increases LC3 and Beclin-1 activity [160]. Santoro et al. show a dose-dependent effect of glycyrrhizin on DA neurodegeneration in the MPTP mouse model [161]. Trehalose may be the most exciting autophagy enhancer, targeting the TFEB pathway by dephosphorylating TFEB and increasing its nuclear translocation [162]. A 2022 study in a rotenone-induced PD mouse model shows the efficient action of trehalose treatment during the prodromal phase, resulting in improved non-motor parameters, motor performances, and the number of DA neurons of SN, but also decreased deposit of brain α-syn compared to non-treated animals [163]. Similar results were corroborated in a non-human primate model overexpressing the A53T α-syn [164]. Trehalose efficiency in autophagy and pathological condition enhancement is broadly more substantial when combined with sodium butyrate, as shown by Kakoty and colleagues in a PFF-rat model orally treated with 2 g/kg of trehalose and 150 mg/kg of sodium butyrate [165]. Cunha and colleagues report nanoparticles and nucleolipidic constructions of approximately 150 nm diameter, carrying trehalose, as safe and enhancing autophagy in vitro [166].

### 5.2. Gene Therapies

Targeting key elements of ALP by gene therapy is another promising avenue that is also being explored. First, evidence of LAMP2A overexpression in human mammary adenocarcinoma cell lines permitted enhanced CMA activity and increased cell survival [167]. One year later, the same results were observed in rats overexpressing α-syn in SN, with an injection of recombinant adenoviruses overexpressing LAMP2A, where amelioration of CMA mechanism, lower α-syn levels, and decrease of DA neurodegeneration [168]. A recent result reported the preventive effect of LAMP2A overexpression in the *SNCA*-induced fly model [169]. Viral-mediated overexpression of Beclin-1 has also proved to have therapeutic interest in rodent models overexpressing human α-syn. Beclin-1 overexpression is also linked to reduced apoptosis and enhanced autophagy [170]. Following Beclin-1 lentiviral injection in the temporal cortex and hippocampus of α-syn transgenic mice, a decrease in α-syn levels was observed through autophagy activation [171]. Finally, transcription factor EB (TFEB) has emerged as a master activator of the ALP machinery, promoting lysosomal biogenesis and enhancing autophagy [172]. Gene therapy using adeno-associated viral (AAV)-mediated TFEB overexpression in wild-type α-syn and human mutated A53T α-syn overexpressing rat models conferred a protective effect on rat midbrain neurons, associated with increased clearance of pathologic α-syn through autophagic recovery [173,174]. In the same study, Beclin-1 gene therapy overexpression shows similar results [174]. While TFEB neuronal expression was sufficient to prevent neurodegeneration in PD models, TFEB oligodendroglial overexpression leads to neuroprotective effects in the transgenic PLP α-syn mouse, as an MSA model [173]. Gene therapies appear as a unique and interesting treatment technology that may be patient-specific by regulating gene expression of specific genotype abnormalities in PD patients, and having a target-specific property depending on the serology type of the adeno-associated virus used is an important point to consider to avoid off-target gene regulation.

### 5.3. Biotechnological Strategy

Recently, new technologies to restore ALP have emerged, particularly by targeting lysosomal pH. Lysosomal acidification, which is defective in neurodegenerative diseases such as Alzheimer’s disease [175,176] and PD [177], is also essential for lysosomal function. Nanocarriers like nanoparticles or nucleolipids encapsulating drugs of interest are one of them [178]. Biocompatible organic acid-loaded nanoparticles (i.e., poly(DL-lactide-co-glycolide) (PLGA) or succinic diacid) targeting restoration of lysosome acidification have been tested and showed promising results in different pathological contexts [177,179,180,181,182,183,184,185,186]. Thus, in vitro studies in several models of lysosomal impairment, including ATP13A2-mutant cells, GBA-mutant fibroblasts from PD patients, and neuroblastoma cells treated with lysosomotropic agents pathologically enhancing lysosomal pH, demonstrate that PLGA acidic nanoparticles can enhance cell survival and restore the physiological pH of the lysosome lumen [177].

Furthermore, those acid-loaded nanoparticles are neuroprotective in two different mouse models of PD: the MPTP mouse model and the LB-seeding mouse model of PD. The same PLGA acidic nanoparticles were assessed in an atherosclerosis mouse model *apoe*^-/-^ reporting a significant increase in lysosome and macrophage numbers and the rescue of lysosomal acidity in macrophages [187]. Strategies enhancing or restoring lysosomal-mediated degradation thus appear as tantalizing neuroprotective/disease-modifying therapeutic strategies. Restoration of functional lysosomes would be of significant therapeutic interest for PD, in which nanotechnologies could play a key role because of their interesting properties to easily cross the BBB and because they may be target-specific to avoid adverse effects and deliver active compounds directly at the desired point.

### 5.4. Drugs Currently in Clinical Trials

In recent years, multiple treatments to restore autophagy have entered clinical testing phases [188]. Ambroxol may be a promising one; firstly, used as a cough syrup, it can enhance GCase activity, permit better lysosomal function, and decrease reactive oxygen species (ROS) apparition [189]. A clinical phase II study is ongoing until 2023 in a single-center, double-blind, randomized, and placebo-controlled manner (NCT02914366) [190]. Based on the promising results of the AIM-PD phase 2 clinical trial (NCT02941822), the University College London is starting a phase clinical trial in 2023, gathering 330 people with PD across approximately 12 clinical centers in the United Kingdom [191]. Another GCase activator drug is in the clinical phase study under the LTI-291 name. This small molecule forms an active complex with GCase and has shown 130% increased activity of GCase in vitro. Phase I study performed in 2021 shows good tolerability and no adverse effects of LTI-291 administration [192]. Venglustat is also a drug that participates in GCase activity amelioration by inhibiting glucosylceramide synthase. Phases I and II performed by Peterschimtt and colleagues resulted in good safety of oral administration [193] and improved glucosylceramide levels in *GBA*-PD patients [194], without observation of severe adverse events.

As described previously, mTOR inhibition through rapamycin capacities can be an interesting strategy for Sirolimus medication. This drug, clinically tried in multiple system atrophy patients, does not show any benefit on pathological rating scales, neuroimaging, or blood biomarkers compared to controls, and adverse events were more frequent for treated patients [195]. Currently, no clinical trial with Sirolimus is planned for PD patients, but it opened the way for mTOR inhibition through rapamycin in neurodegenerative disorders.

Nilotinib or Tasigna^®^ drug is a tyrosine kinase inhibitor enhancing autophagy activity [196], showing interesting results on the reduction, with a rate twice as low, of PD hallmark α-syn in the A53T mouse model [197]. Tested on clinical phase I and phase II studies, it proves its safety and tolerability, with no adverse effects observed and a decrease in oligomeric forms of the α-syn in PD patients [198,199]. However, a recent meta-analysis concluded that it had no effect on motor outcomes [200]. Pagan and colleagues support starting a phase 3 trial project to evaluate nilotinib 300 mg in a more extensive multicenter study [199].

## 6. Conclusions

At the same time, the growing knowledge of PD, the ALP mechanisms, and the link between this disease and this important cellular mechanism has allowed for the generation of animal models, promising new research and knowledge. Presently, multiple potential treatments to restore the critical mechanism of autophagy in PD are being clinically tested [188] and will soon provide answers on their capacity to act as disease modifiers in a pathology for which there is no curative treatment yet. In addition, the future should see the development of drugs targeting the mTOR pathway or other autophagy modulators, like TFEB and Beclin-1. The strategy of regulating autophagy to treat neurodegenerative disorders is an interesting avenue, with multiple and diverse possible solutions, from drugs to nanotechnologies by way of gene therapies. Of interest, TFEB-based therapy in the AD context has proven beneficial [201,202]. Beyond that, all these diseases have a dysfunction of the ALP pathway in common. As a result, any treatment developed for one of the neurodegenerative diseases may have an impact on the others. The growing interest in and improved knowledge of this mechanism in recent years should accelerate future discoveries, increasing the chances of rapidly seeing possible therapeutics for PD patients in the following years.

## Figures and Tables

**Figure 1 cells-12-00621-f001:**
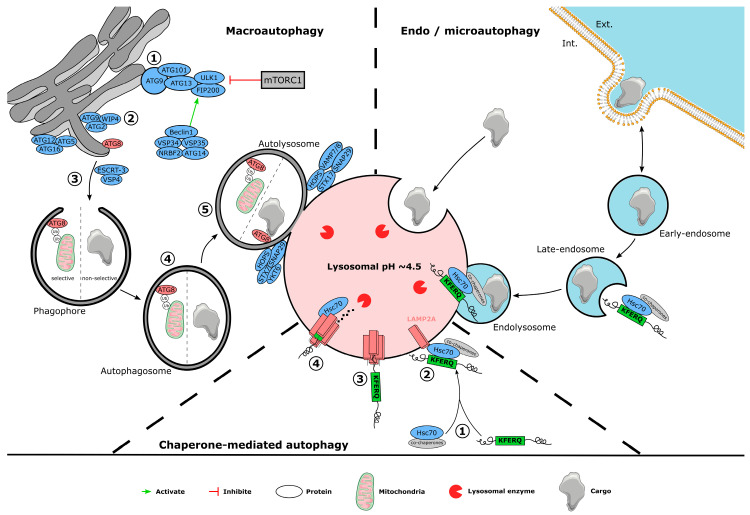
**Schematic representation of different autophagic pathways**. Macroautophagy is composed of (1) the initiation phase, (2) the elongation phase, (3) the separation and enclosure of the phagophore, (4) the maturation step, and (5) the fusion with the lysosome. Chaperone-mediated autophagy starts with (1) the recognition of the KFERQ motif by the Hsc70 complex, (2) leading the cargo to LAMP2A, which (3) translocates, unfolds, and (4) degrades the protein. Endo/microautophagy is characterized by cargo capture directly by the endosome or lysosome before lysosomal degradation.

**Figure 2 cells-12-00621-f002:**
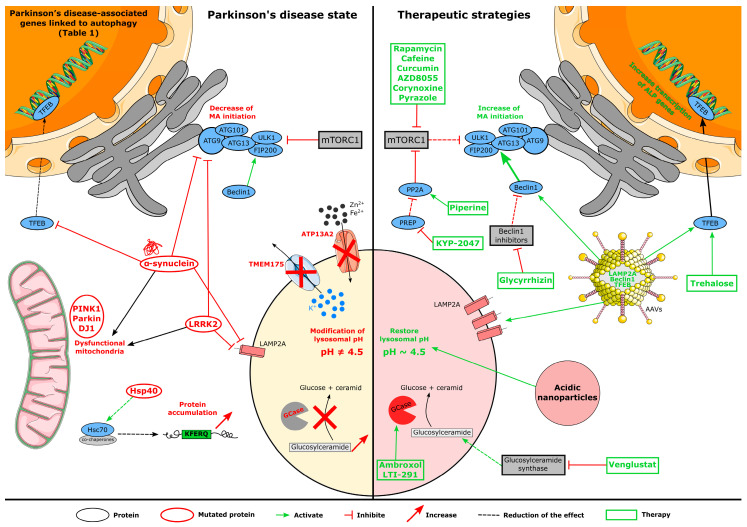
Schematic representation of the impact of mutated proteins on the autophagy mechanism (**left panel**) and the targets of different treatments on autophagy (**right panel**).

**Table 1 cells-12-00621-t001:** **Summary of familial PD-associated genes.** This table gathers the names of genes for which mutations are reported in familial PD cases, their associated protein name, their inheritance type (AD: autosomal dominant; AR: autosomal recessive; XL-D: X-linked dominant), and their involvement in LB hallmark and ALP.

Gene	Protein	Inheritance	LB	ALP Involvement
*SNCA*	α-synuclein	AD	Yes [5]	Yes [24]
*LRRK2*	Leucine-rich repeat kinase 2	AD/AR	Yes [25]	Yes [26]
*VPS35*	Vacuolar protein sorting ortholog 35	AD	Unknown	Yes [27]
*ATXN2*	Ataxin 2	AD	Yes [28]	Yes [29]
*GCH1*	GTP cyckihydrolase 1	AD	No [30]	Unknown
*PRKN*	Parkin RBR E3 ubiquitin protein ligase	AR	No [31]	Yes [32]
*PINK1*	PTEN-induced putative kinase 1	AR	Yes [33]	Yes [34]
*PARK7*	Parkinsonism-associated deglycase/DJ1	AR	Yes [35]	Yes [36]
*ATP13A2*	ATPase cation transporting 13A2	AR	No [37]	Yes [38]
*DCTN1*	Dynactin subunit 1	AD	No/few [39]	Yes [40]
*DNAJC6*	DnaJ Heat Shock Protein Family (Hsp40) Member C6	AR	Unknown	Yes [41]
*DNAJC13*	DnaJ Heat Shock Protein Family (Hsp40) Member C13	AD	Yes [42]	Yes [42]
*EIF4G1*	Eukaryotic translation initiation factor 4G1	AD	Yes [43]	Yes [44]
*FBXO7*	F-Box Protein 7	AR	Unknown	Yes [45]
*HTRA2*	HTRA serine peptidase 2	AD	Yes [46]	Yes [47]
*PLA2G6*	Phospholipase A2 group 6	AR	Yes [48]	Yes [49]
*SYNJ1*	Synaptojanin 1	AR	Unknown	Yes [50]
*SPG11*	Spatacsin	AR	Yes [51]	Yes [52]
*CHCHD2*	Coiled-Coil-Helix-Coiled-Coil-Helix Domain Containing 2	AD	Yes [53]	Yes [53]
*LRP10*	LDL Receptor-Related Protein 10	AD	Yes [54]	Unknown
*RAB39B*	Ras-related protein Rab-39B	XL-D	Yes [55]	Yes [56]
*TAF1*	TATA-box binding protein associated factor 1	XL-D	No [57]	Unknown
*TMEM230*	Transmembrane Protein 230	AD	Yes [58]	Yes [59]
*UQCRC1*	Ubiquinol-Cytochrome C Reductase Core Protein 1	AD	Unknown	Unknown
*VPS13C*	Vacuolar Protein Sorting 13 Homolog C	AR	Yes [60]	Yes [60]
*TMEM175*	Endosomal/lysosomal proton channel TMEM175	Unknown	Yes [61]	Yes [62]

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
