# Peer review of "Pathogenic Aspects and Therapeutic Avenues of Autophagy in Parkinson’s Disease"

_cells, 2023, doi:10.3390/cells12040621_

Round 1

Reviewer 1 Report

The manuscript by Kinet and Dehay aimed to discuss Parkinson's disease features, the autophagy mechanisms, and their involvement in pathogenesis, like the potential treatments in the preclinical and 14 clinical phase studies.

The review is well-written and structured, and the reference list covers the relevant literature adequately and includes recent developments.  However, several issues must be addressed before the manuscript is suitable for publication in the Journal Cells.

Minor comments:

The author should review reference 59 because the title needs to be corrected.

References 119 and 120 are repeated.

The authors should improve the abstract to highlight advances in knowledge and the use of models for the search for therapeutic alternatives in PD.

Major comments:

The authors should include in section 1.2 a figure explaining the general mechanisms of autophagy signaling (macroautophagy, mitophagy, and autophagy) in dopaminergic neurons and the cellular phenotypic associated.

In section 3, the authors should include another figure to show the therapeutic targets of the drugs that regulate macroautophagy, mitophagy, and autophagy described in the review. 

The lines 164 to 173, this reviewer recommends discussing the role of TMEM175 as a growth-factor-activated lysosomal K+ channel (see ref: A growth-factor-activated lysosomal K+ channel regulates Parkinson's pathology | Nature).

 In line 279, the authors should clarify the cellular model used in this study (Ref 162).

This reviewer recommends discussing a recent clinical trial using Sirolimus (see ref: mTOR Inhibition with Sirolimus in Multiple System Atrophy: A Randomized, Double-Blind, Placebo-Controlled Futility Trial and 1-Year Biomarker Longitudinal Analysis - PubMed (nih.gov)).

The authors should include the limitations in pharmacological, genetic, and biotechnological treatments and possible side effects.

Author Response

Review Report (Reviewer 1)

The manuscript by Kinet and Dehay aimed to discuss Parkinson's disease features, the autophagy mechanisms, and their involvement in pathogenesis, like the potential treatments in the preclinical and 14 clinical phase studies.

The review is well-written and structured, and the reference list covers the relevant literature adequately and includes recent developments.  However, several issues must be addressed before the manuscript is suitable for publication in the Journal Cells.

Minor comments:

The author should review reference 59 because the title needs to be corrected.

To our knowledge, the reference “Scherz-Shouval, R. and Z. Elazar, ROS, mitochondria and the regulation of autophagy.Trends Cell Biol, 2007. 17(9): p. 422-7.” does not contain any error (now listed as reference 65)

References 119 and 120 are repeated.

References 119 and 120 share a similar title and the same first author but aren’t the same reference. To avoid confusion and use the princeps article, the references were modified.

The authors should improve the abstract to highlight advances in knowledge and the use of models for the search for therapeutic alternatives in PD.

We thank the reviewer for his/her suggestion. We have now improved the Abstract in the revised version to highlight the recent advances in knowledge of Parkinson’s disease and autophagy.

Major comments:

The authors should include in section 1.2 a figure explaining the general mechanisms of autophagy signaling (macroautophagy, mitophagy, and autophagy) in dopaminergic neurons and the cellular phenotypic associated.

We thank the reviewer for his/her suggestion. A new Figure (Figure 1) has been added to describe the general mechanisms of autophagy signaling (macroautophagy, mitophagy, and autophagy).

In section 3, the authors should include another figure to show the therapeutic targets of the drugs that regulate macroautophagy, mitophagy, and autophagy described in the review.

We thank the reviewer for his/her suggestion. A second new Figure (Figure 2) has been added to summarize the different treatments addressed in this manuscript.

The lines 164 to 173, this reviewer recommends discussing the role of TMEM175 as a growth-factor-activated lysosomal K+ channel (see ref: A growth-factor-activated lysosomal K+ channel regulates Parkinson's pathology | Nature).

The sentence has been modified to highlight the grow-factor activated function of TMEM175: “Indeed, this channel regulating lysosomal functions through proton-activating, or growth-factor activating manner” (lines 198-199)

In line 279, the authors should clarify the cellular model used in this study (Ref 162).

The cellular model in reference 162 was added: “human mammary adenocarcinoma cell lines” (lines 326-327).

This reviewer recommends discussing a recent clinical trial using Sirolimus (see ref: mTOR Inhibition with Sirolimus in Multiple System Atrophy: A Randomized, Double-Blind, Placebo-Controlled Futility Trial and 1-Year Biomarker Longitudinal Analysis - PubMed (nih.gov)).

We thank the reviewer for his/her suggestion. Sentences treating the clinical trial of Sirolimus were added in lines 391-397: “As described previously, mTOR inhibition through rapamycin capacities can be an interesting strategy tried by Sirolimus medication. This drug, clinically tried in multiple system atrophy patients, does not show any benefit on the pathological rating scale, neuroimaging, or blood biomarkers compared to control, and adverse events were more frequent for treated patients. Currently, no clinical trial with Sirolimus is planned for PD patients, but it opened the way for mTOR inhibition through rapamycin in neurodegenerative disorders.”

The authors should include the limitations in pharmacological, genetic, and biotechnological treatments and possible side effects.

We thank the reviewer for his/her suggestion. We have now added the limitations and possible side effects in the revised manuscript in lines 293-296, 345-349, and 369-372.

Reviewer 2 Report

In the current submission, authors aimed to provide a detailed investigation on Parkinson’s disease features, the autophagy mechanisms and their involvement in pathogenesis. In addition, authors have further highlighted current methods to restore this cellular pathway. Despite interesting review, authors must address following comments during revision:

1.      Introduction should be separately written by highlighting background information, research gap, rationale and motivation to the review.

2.      What will be the benefit of the study? The introduction should specifically indicate the significance of the study as a separate paragraph. Moreover, this will inspire readers to take an interest in the subject matter. Please provide a structured ending statement in the introduction section that why this review article is important considering the field, its significance and novelty. What makes this review article unique? 

3.      To write a good review, at least 2-3 figures are mandatory.

4.      A critical assessment of the present knowledge with some clear conclusions what all these results mean, and directions for future research and potential applications should be strengthened.

5.      The authors are recommended to compare the discussed therapeutic modalities with traditional therapeutic approaches such as small-molecule inhibitors and gene-based strategies, etc.

6.      More emphasis and in-depth discussion of the fundamental as well as clinical prospects of each intervention are needed.

7.      The authors should discus about recently approved drugs for the patients with Parkinson’s disease. In addition, how these drugs work should be justified. They should briefly describe their functioning as well.

8.      The authors should briefly describe Brain organoid technology in Parkinson’s disease.

9.      It would be better to compare the status of clinical applications of these technologies in Alzheimer’s and Parkinson’s disease.

10.  Perspectives and future directions should be crisp.

11.  There are grammatical mistakes and typographical errors in the manuscript. The author should recheck this manuscript carefully and remove all such errors.

12.   Many statements have been made without any proper justifications or citations.

13.  All references should be thoroughly checked especially authors must confirm only the relevant publication should be cited.

Author Response

Review Report (Reviewer 2)

In the current submission, authors aimed to provide a detailed investigation on Parkinson’s disease features, the autophagy mechanisms and their involvement in pathogenesis. In addition, authors have further highlighted current methods to restore this cellular pathway. Despite interesting review, authors must address following comments during revision:

  1. Introduction should be separately written by highlighting background information, research gap, rationale and motivation to the review.

The Introduction section has been moved to a dedicated section to satisfy the reviewer in the revised version.

  1. What will be the benefit of the study? The introduction should specifically indicate the significance of the study as a separate paragraph. Moreover, this will inspire readers to take an interest in the subject matter. Please provide a structured ending statement in the introduction section that why this review article is important considering the field, its significance and novelty. What makes this review article unique?

We thank the reviewer for his/her suggestion. The introduction section has been written completely to satisfy the reviewer (lines 23-44).

  1. To write a good review, at least 2-3 figures are mandatory.

We thank the reviewer for his/her suggestion. We added two new Figures, the first one describing the general mechanisms of autophagy and the second one summarizing the different treatments addressed in the manuscript.

  1. A critical assessment of the present knowledge with some clear conclusions about what all these results mean and directions for future research and potential applications should be strengthened.

We added some sentences to strengthen this aspect in lines 293-296, 345-349, and 369-372.

  1. The authors are recommended to compare the discussed therapeutic modalities with traditional therapeutic approaches such as small-molecule inhibitors and gene-based strategies, etc.

We added some sentences to strengthen this aspect in lines 293-296, 345-349, and 369-372.

  1. More emphasis and in-depth discussion of the fundamental as well as clinical prospects of each intervention are needed.

We discussed this aspect in a revised version of section 5.4.

  1. The authors should discuss about recently approved drugs for the patients with Parkinson’s disease. In addition, how these drugs work should be justified. They should briefly describe their functioning as well.

We thank the reviewer for his/her suggestion. This is not clear to us what are the recently approved drugs for patients with Parkinson’s disease the reviewer meant. We tried to make an update on all the ongoing clinical trials, including 2023.

  1. The authors should briefly describe Brain organoid technology in Parkinson’s disease.

We have added in lines 256-261: “The increasing number of animal models linked to ALP dysfunction to mimic PD phenotype is a key step for understanding this pathology and developing and testing possible future treatments. In addition to these latest findings, it is important to notice the usefulness of non-animal models, and in particular brain organoids in the first steps of the study, due to their capacity to generate complex and multicellular systems from patients' pluripotent stem cells”

  1. It would be better to compare the status of clinical applications of these technologies in Alzheimer’s and Parkinson’s disease.

We added some sentences to cover this aspect in conclusion.

  1. Perspectives and future directions should be crisp.

We thank the reviewer for his/her suggestion. The Perspectives and future directions in the conclusion section have been developed.

  1. There are grammatical mistakes and typographical errors in the manuscript. The author should recheck this manuscript carefully and remove all such errors.

The grammatical and typographical aspects of the article have been verified throughout the manuscript.

  1. Many statements have been made without any proper justifications or citations.

We corrected this aspect throughout the manuscript.

  1. All references should be thoroughly checked, especially authors must confirm only the relevant publication should be cited.

We replaced some references to cite the relevant publications, keeping in mind that the number of references accepted constrains us.

Reviewer 3 Report

The article is very good article, well-written, and relevant to the field of Parkinson's Disease and Autophagy. Great topics were mentioned and well described. 

The review brings what is the most up-to-date about the autophagy mechanisms and their involvement in the pathogenesis of Parkinson's Disease. It also discusses methods to restore this cellular pathway and the potentially curative treatments in preclinical and clinical phase studies.

This is a very relevant topic to the field, and several potential therapies are based on that. So, it is good to bring the teme up and discuss it more.

There are a few other review articles published in the same field. However, the author of this manuscript made better work by getting more descriptions of genes related to the disease.

the conclusion of the review answers the question addressed in the article.

There is only one table in the review showing genes related to autophagy and Parkinson's Disease.

Author Response

Review Report (Reviewer 3)

The article is very good article, well-written, and relevant to the field of Parkinson's Disease and Autophagy. Great topics were mentioned and well described. The review brings what is the most up-to-date about the autophagy mechanisms and their involvement in the pathogenesis of Parkinson's Disease. It also discusses methods to restore this cellular pathway and the potentially curative treatments in preclinical and clinical phase studies.

We thank the reviewer for such a positive appraisal of our work.

This is a very relevant topic to the field, and several potential therapies are based on that. So, it is good to bring the teme up and discuss it more.

The chapter describing the different therapeutic approaches was extended.

There are a few other review articles published in the same field. However, the author of this manuscript made better work by getting more descriptions of genes related to the disease. The conclusion of the review answers the question addressed in the article. There is only one table in the review showing genes related to autophagy and Parkinson's Disease.

Two figures have been added to the review as well as the Table.

Round 2

Reviewer 2 Report

Revised manuscript is acceptable now.